# PD-L1 Status in Gastric Cancers, Association with the Transcriptional, Growth Factors, AKT/mTOR Components Change, and Autophagy Initiation

**DOI:** 10.3390/ijms222011176

**Published:** 2021-10-16

**Authors:** Liudmila Spirina, Alexandra Avgustinovich, Sergei Afanas’ev, Maxim Volkov, Alexey Dobrodeev, Olga Cheremisina, Dmitry Kostromitsky

**Affiliations:** 1Cancer Research Institute, Tomsk National Research Medical Center, 634000 Tomsk, Russia; aov862@yandex.ru (A.A.); AfanasievSG@oncology.tomsk.ru (S.A.); dok75-75@mail.ru (M.V.); dobrodeev@oncology.tomsk.ru (A.D.); cheremisinaov@oncology.tomsk.ru (O.C.); info@oncology.tomsk.ru (D.K.); 2Biochemistry and Molecular Biology Department, Medical and Biological Faculty, Siberian State Medical University, 634000 Tomsk, Russia

**Keywords:** PD-L1, gastric cancers, transcriptional factors, growth factors, AKT/mTOR component, LC3B

## Abstract

Introduction: The programmed death receptor ligand 1 (PD-L1) immunohistochemistry (IHC) assay is a widely used selection method for pembrolizumab treatment in gastric cancer (GC) patients. PD-L1 is the main regulator of immunity in oncogenesis. Material and methods: The study included 38 patients with GC. The combined treatment consisted of neoadjuvant FOLFOX6, or FLOT, chemotherapy and surgery. PD-L1 + tumor status was recorded in 12 patients (CPS > 5), with a negative status recorded in 26 patients. RT-PCR determined the expression of molecular markers. The level of LC3B protein was detected by Western Blotting analysis. Results: An overexpression of PD-1, PD-L2 in the tumor is associated with AKT/mTOR mRNA profile change and autophagy initiation in IHC PD-L1 positive GCs. NACT influences these biological features, modifying the expression of AKT/mTOR components and autophagic flux. In PD-L1 positive cancers, the effect of NACT and molecular markers rearrangements are essential compared to the PD-L1 negative cancers. Conclusion: The IHC PD-L1 status in gastric cancers is the significant marker of cancer progression, recovering the multiple inner mechanisms of cancer spreading and leading to ineffective therapy. Autophagy induction and angiogenesis are found in PD-L1 positive gastric cancers.

## 1. Introduction

In recent decades, gastric cancers have remained a global burden. The fundamental issue in modern oncology is the search for effective anti-cancer drugs and neoadjuvant chemotherapy (NACT) regimens. The expression of programmed death receptor ligand 1 (PD-L1) can be determined by an immunohistochemistry (IHC) assay, a widely used selection method for pembrolizumab treatment in gastric cancer (GC) patients. PD-L1 expression in GC patients is correlated with molecular features. In 59.3% of GC patients with a PD-L1 positive status, MSI (microsatellite instability) and EBV (Epstein–Barr virus) positivity was found [1]. A limited number of patients achieved clinical benefits, highlighting the importance of a greater selection of patients or the need for additional treatment to overcome this resistance to PD-1/PD-L1 blockade [2].

Tumor infiltration by immunocompetent cells and, consequently, the production of PD receptors, PD-L1, and PD-L2 ligands by tumor cells, are also controlled by AKT/mTOR kinases [3]. An increase in PD-L1 and PD-L2 expression is associated with cancer spreading and the involvement of distant organs.

Multiple molecular mechanisms are involved in GCs immunity. PD-L1 expression in the residual tumor can be used as a prognostic marker in patients after receiving NACT [4]. High CTLA-4 and p-Akt expression levels in pretreatment tumor cells were associated with poor clinical outcomes [5]. Positive correlations were found among PD-L1, c-MET, and HGF, based on TCGA datasheets and paired cancer specimens. Moreover, the resistant mechanisms increase PD-L1 expression and attenuate the activation and cytotoxicity of lymphocytes [6].

Increased AKT mRNA expression correlates with lymph node metastasis, while decreased PTEN mRNA levels correlate with advanced tumor stage and lymph node metastasis [7]. The findings reveal that PD-L1 expression promotes the EMT of cancer cells resistant to anti-cancer therapy [8].

Transcriptional and growth factors are also implemented during oncogenesis via a modification of the immune response. Hypoxia-inducible factor 1 (HIF-1) plays an indispensable role in the hypoxic tumor microenvironment [9].

Furthermore, PD-L1 positive immune cells also initiate autophagy—the most potent mechanism of cancer progression. The pattern of proteins affecting immunity includes autophagic proteins. It is known the PD-L1 positive cells express LC3B, suggesting the need for further investigation into the modulation of the immune microenvironment [10]. However, misfolded proteins, damaged mitochondria, and other unwanted components in cells can be decomposed and reused via autophagy in some specific cases (including hypoxic stress, low energy states, or nutrient deprivation) [11]. Dynamic LC3B and p62 changes are found in gastric tumorigenesis and are used as predictive biomarkers, and potential therapeutic targets, for GCs [12].

The effects of NACT on immune markers and the biological features in cancers remain largely unknown. NACT significantly alters the expression of immunosuppressive molecules, suggesting a choice of immune response combinations [13]. NACT increases immune infiltration. The evaluation of immune parameters in post-NACT tumors may help select patients for immunotherapy trials [14].

Moreover, the effect of NACT on tumor-infiltrating lymphocytes and PD-L1 expression in tumors has clinical significance [15]. AKT/mTOR activation is a critical event in response to NACT [16]. Commonly, neoadjuvant PI3K/mTOR/AKT inhibition reduces tumor growth [17]. The cellular degradation process has a complex role in tumorigenesis and resistance to cancer treatment in humans; therefore, a mechanism is required to ensure autophagy regulation. However, the high expression of autophagy-related proteins post-treatment shows a strong, negative association with the efficacy of NACT in cancer patients [18].

Autophagy may contribute to an acquired resistance against Her2-targeted therapy in cancers; therefore, combining Her2 and autophagy inhibition might be beneficial for cancer patients [19]. In addition, the anti-cancer treatment can also be seen to reduce the expression of mTOR, p62, BCL2, and upregulated Beclin 1 and LC3-I/II, which are significant autophagy-related genes. These processes induce potent cell apoptosis in cancer cells. NACT exerts antitumor activities by blocking molecular signaling pathways [20]. Autophagy regulates PD-L1 expression in gastric cancer through the p62/SQSTM1-NF-κB pathway. Thus, the pharmacological modulation of autophagy may influence the therapeutic efficacy of PD-L1 blockade in gastric cancer [21].

The prognostic significance of autophagy-related proteins highlights the importance of autophagy in the biologic behavior of chemoresistant cancer cells. LC3B can be used as a prognostic marker in cancer patients after NACT. Furthermore, evaluating and targeting autophagy in a neoadjuvant setting may help prevent disease relapse in patients [22]. This study aimed to analyze the PD-1, PD-L1, PD-L2, transcriptional, growth factors, and LC3B expression in GC patients associated with the IHC PD-L1 status.

## 2. Results

### 2.1. PD-Status in Cancers Impacts Transcriptional Growth Factors, AKT/mTOR Components, and PD-1, PD-L1, and PD-L2 Expression

Table 1 summarizes the PD-1, PD-L1, PD-L2 expression in IHC PD-L1 negative and positive cancers. PD-1 and PD-L2 expression increased by 3.86 and 2.43 times in PD-L1 positive GCs.

The tumor infiltration by immunocompetent cells and, consequently, the production of PD receptors and PD-L1, PD-L2 ligands by tumor cells are also controlled by an AKT/mTOR signaling cascade, transcription, and growth factors [3]. The NF-κB p50 and HIF-2 expression increased by 15.05 and 12.34 times in PD+ tumors compared to the negative ones (Table 2).

### 2.2. PD-Status in Cancers and Autophagy Initiation

LC3B mRNA level was associated with PD-L1 tumor status. We found that mRNA and protein content increases by 3.49 and 1.76 times in IHC PD-L1 positive cancers compared to negative ones (Table 3, Figure 1).

### 2.3. Molecular Mechanisms of Modified Immunity Are Due to the Impact of NACT, the Role of the AKT/mTOR Signaling Cascade, and LC3B

Previous studies have noted the changes in the expression of the AKT/mTOR signaling cascade associated with GC extension [3]. After the NACT, we revealed a decrease in 4EBP1 expression by 2.2 times in the tumors (Table 4), with an increase in VEGF expression and CAIX by 7.25 and 5.1 times, compared to the non-altered tissues.

We found an LC3B expression increase of 5.12 times after NACT (Table 5). However, the PD, PD-L1, and PD-L2 expression in cancers before and after NACT did not change.

We detected a decrease in AKT and mTOR mRNA levels in 5.66 and 11.05 times in PD-L1 positive GCs with the enhancement of the HIF-1, VEGF, and CAIX, increasing by 26.8, 8.0, and 14.19 times compared to the negative ones (Table 6). The obtained change in the biological properties of cancers highlights the involvement of multiple inner mechanisms in the regulation behavior of cancer cells.

Table 7 presents data on the growth of LC3B protein levels in GCs after NACT. The adaptation to the anti-cancer therapy was accompanied by autophagy induction. We indicated the PD-1 expression enhancement in IHC PD-L1 positive tumors. A 24.1 fold increase of the indicator shows an immunity modification depending on the PD-L1 status in GCs.

## 3. Discussion

The PD-L1 tumor status is closely associated with increased PD-1 and PD-L2 expression. It has been determined that multiple molecular mechanisms are involved in the GCs immunity [5]. Positive correlations were found among PD-L1, growth, and nuclear factors in cancer cells [6]. Moreover, an increase in inflammatory factors (NF-κB and HIF-2) was detected in PD-L1+ GCs. Therefore, immunity modulation is an essential oncogenic trigger [15].

We revealed that, after the treatment, a decrease in the AKT/mTOR signaling cascade was more pronounced in IHC PD-L1 positive cancers. Hypoxia and HIF signaling were found to activate cancer progression due to the vascular endothelial growth factor, prostaglandin E2, and PD-1 overexpression [9,23].

We revealed molecular markers and autophagy-related proteins modification in GCs. The decrease in 4EBP expression and LC3B mRNA level growth is crucial in GCs, involving specific hallmarks in cancer cells.

Autophagy, a cellular degradation process, has complex roles in tumorigenesis and determines the resistance to anti-cancer treatment in humans. An increase in LC3B content recovers the inner mechanism of PD-L1 overproduction in GCs. The expression of autophagy-related proteins shows a strong negative association with the response to NACT in patients [18]. This study found that the fundamental explanation for GC aggressiveness depends on immunity modulation and pre-existing cellular resistance. Autophagy is an essential process in tumor behavior, especially in IHC PD-L1 positive GCs. Consequently, changes to this behavior may be associated with the varying influence of NACT.

The findings revealed that PD-L1 promotes GC resistance to anti-cancer therapy [8], noting an increased PD-1 expression in cancers after the NACT. Currently, there is only isolated, ambiguous information regarding the relationship between molecular markers and the effects of NACT. It is believed that primary resistance to anti-cancer therapy is a multistep process [9]. NACT gastric cancer therapy affects the 4EBP1 expression in the tumor. A high expression of AMPK, mTOR, and 4EBP1 in tumors before treatment mediates the NACT effect in GC patients and is associated with tumor response to anti-cancer therapy [3]. The impact of NACT on cancers depends on immunity, PD-1 expression, and PD-L1 protein content. The activation of autophagy, and the overexpression of transcriptional and growth factors, are more pronounced in GCs with PD-L1 positive status.

NACT is the most potent factor, changing the immune response and affecting most molecular processes. It is known that NACT exerts antitumor activities and affects the autophagy pathway [20]. LC3B levels can be used as a prognostic marker in patients after they have received NACT, highlighting the importance of autophagy in the biologic behavior of chemoresistant cancer cells. Furthermore, evaluating and targeting autophagy in a neoadjuvant setting may help to prevent disease relapse in patients [22]. Similarly, PD-L1 positive immune cells also initiate autophagy [19]. The pattern of proteins affecting immunity includes autophagic proteins. It is known that the PD-L1 positive cells express LC3B, suggesting the need for further investigations regarding autophagy and the immune microenvironment [10,11,12].

Consequently, NACT significantly alters the expression of molecular markers determining the modified invasive and metastatic potential in cancers. NACT increases immune infiltration and PD-L1 overexpression. The marker levels are a prognostic at diagnosis and remain a prognostic after NACT. Evaluating immune parameters in the post-NACT tumor may help to select patients for immunotherapy trials. Moreover, the effect of NACT on biological processes and oncogenesis is understood. AKT activation is an important event in response to NACT and affects the body’s sensitivity to anti-cancer therapy. The data explains that the growth in LC3B expression and protein content in IHC PD-L1 positive cancers makes them the most resistant to NACT. A focus on autophagy as a therapeutic target could therefore improve the efficacy of NACT.

## 4. Materials and Methods

The study included 38 patients with GC. Combined GC therapy consisted of neoadjuvant FOLFOX6, or FLOT, chemotherapy and surgery. The patients underwent eight courses of NACT with 14 days of recess. The time between the end of the last NACT course and surgery was from 4 to 8 weeks. Stage T3N0-2M0 and T4N0-2M0 were revealed in 10 and 28 patients, respectively. Stage T1-4N0M0 was diagnosed in 14 patients, T1-4N1M0-in 16 patients, and T1-4N2M0— in 8 patients (Table 8). PD-L1 + tumor status was recorded in 12 patients (CPS > 5), with a negative status in 26 patients. The additional clinical characteristics are presented in Table 8.

In the presence of positive HER2 tumor status, trastuzumab 4 mg/kg was administered intravenously on the first day of chemotherapy (loading dose 6 mg/kg), with a PD-L1-positive status pembrolizumab 200 mg IV drip once every 3 weeks. The combined effect of the treatment was evaluated in patients with locally advanced GC and was carried out using the RECIST 1.1 scale.

The Ethics Committee approved the study of the Cancer Research Institute of the Tomsk National Research Medical Center. The material used within this study (tumor tissue and normal stomach tissue located at a distance of at least 1 cm from the border of the tumor) was obtained during diagnostic video gastroscopy. The collected specimens were frozen and stored at −80 °C. To analyze the expression of molecular markers and RNA isolation, the tissue samples were placed in an RNAlater solution (Ambion) and held at −80 °C after a 24 h incubation at 4 °C.

HER2 status detection. A combination of IHC and fluorescence in situ hybridization (FISH) was used for HER-2 status detection. HER-2 (+3) was assessed as positive. If the results of IHC were ambiguous (2+), the FISH technique was used.

IHC PD-L1 status detection. IHC was applied for the detection of the PD-L1 status. PD-L1 + tumor status was detected in tissues with CPS >5.

IHC procedure. Paraffin-embedded tissue sections were deparaffinized in xylene for 15 min, rehydrated through a decreasing series of graded ethanol (100, 90, 80, and 70%), incubated in 3% hydrogen peroxide/methanol solution for 30 min at room temperature to quench the endogenous peroxidase activity, and washed with distilled water for 5 min. For PD-L1 detection, antigen retrieval was performed by heating sections with a 0.01 mol/L citrate buffer (pH 6.0) at 100 °C for 3 min in a microwave. Following antigen retrieval, the sections were blocked with 10% normal goat serum (Vector Laboratories, Inc., Burlingame, CA, USA) for 20 min at room temperature in a humidified chamber. The sections were subsequently incubated with the following primary antibodies at 4 °C overnight: anti-PD-L1 (rabbit; 1:500; cat. no. ab205921; Abcam, Göttingen, Germany). Following primary antibody incubation, the sections were incubated with a secondary biotinylated antibody (cat. no. PK-6101, Vector Laboratories, Inc.) using the avidin-biotin complex method. Color development was performed by setting the sections with a 0.02% Histofine^®^ DAB substrate (Nichirei Biosciences, Inc., Tokyo, Japan) at room temperature before counterstaining with Mayer’s hematoxylin. Stained cells were visualized under an optical microscope at a low (×20) and high (×100 and×200) magnification.

RNA extraction. Total RNA was extracted using the RNeasy Mini Kit containing DNase I (Qiagen, Hilden, Germany). The concentration and purity of the isolated RNA were assessed spectrophotometrically on a NanoDrop 2000 spectrophotometer (Thermo Scientific, Waltham, MA, USA). RNA concentration varied from 80 to 250 ng/μL; A260/A280 = 1.95–2.05; A260/A230 = 1.90–2.31. The integrity of the isolated RNA was evaluated using capillary electrophoresis on a TapeStation (Agilent Technologies, Santa Clara, CA, USA) using an R6K ScreenTape kit (Agilent Technologies). RIN was 5.6–7.8. Quantitative reverse transcription PCR was conducted in a reaction mixture (25 μL) containing 12.5 μL BioMaster HS-qPCR SYBR Blue (Biolabmix, Novosibirsk, Russia) and 300 nM of forward and reverse primers. PD-L2: F 5′-GTTCCACATACCTCAAGTCCAA-3′, R 5′-ATAGCACTGTTCACTTCCCTCTT-3′; PD-L1: F 5′-AGGGAGAATGATGGATGTGAA-3′, R 5′-ATCATTCACAACCACACTCACAT-3′; PD-1-1: F 5′-CTGGGCGGTGCTACAACT-3′, R 5′-CTTCTGCCCTTCTCTCTGTCA-3′; 4-BP1: F 5′-CAGCCCTTTCTCCCTCACT-3′, R 5′-TTCCCAAGCACATCAACCT-3′; AKT1: F 5′-CGAGGACGCCAAGGAGA-3′, R 5′-GTCATCTTGGTCAGGTGGTGT-3′; C-RAF: F 5′-TGGTGTGTCCTGCTCCCT-3′, R 5′-ACTGCCTGCTACCTTACTTCCT-3′; GSK3b: F 5′-AGACAAGGACGGCAGCAA-3′, R 5′-TGGAGTAGAAGAAATAACGCAAT-3′; 70S kinase alpha: F 5′-CAGCACAGCAAATCCTCAGA-3′, R 5′-ACACATCTCCCTCTCCACCTT-3′; m-TOR: F 5′-CCAAAGGCAACAAGCGAT-3′, R 5′-TTCACCAAACCGTCTCCAA-3′; PDK1: F 5′-TCACCAGGACAGCCAATACA-3′, R 5′-CTCCTCGGTCACTCATCTTCA-3′; VHL F 5′-GGCAGGCGAATCTCTTGA-3′, R 5′-CTATTTCCTTTACTCAGCACCATT-3′; AMPK: F 5′-AAGATGTCCATTGGATGCACT-3′, R 5′-TGAGGTGTTGAGGAACCAGAT-3′; LC3B: F 5′-CCCAAACCGCAGACACAT-3′, R 5′-ATCCCACCAGCCAGCAC-3′; GAPDH: F 5′-GGAAGTCAGGTGGAGCGA-3′, R 5′-GCAACAATATCCACTTTACCAGA-3′. A preincubation at 94 °C for 10 min was used to activate the Hot Start DNA polymerase and denature the DNA. This was followed by 40 amplification cycles of denaturation at 94° for 10 s and annealing at 60° for 20 s. The primers were selected using Vector NTI Advance 11.5 software and the NCBI database. GADPH served as the reference gene; the expression of each specific mRNA was standardized relative to GADPH expression.

Determination of LC3B content. Electrophoresis SDS-PAGE (Laemmli) was used. The protein was transferred to a 0.2-/xm pore-sized PVDF membrane (GE Healthcare, Chalfont Saint Giles, UK), either at 150 mA or 100 V for 1 h using a Bio-Rad Mini Trans-Blot electrophoresis cell. The membrane was incubated in a 1:2500 dilution of monoclonal mouse anti-human LC3B (Affinity Biosciences, Cincinnati, OH, USA) at 4 °C overnight.

PVDF samples were incubated in an Amersham ECL western blotting detection analysis system (GE Medical Systems Information Technologie, Milwaukee, WY, USA). The results were standardized using the beta-actin expression in a sample and were expressed in percentages to the protein content in non-transformed tissues. The level of protein in normal gastric tissue was indicated as 100.

Statistical analysis. Statistical analysis was performed using SPSS 19.0 software. Data were expressed as median and ranges. The Mann–Whitney test was used for comparing differences in mean values. Nonparametric one-way ANOVA on ranks was carried out to test whether samples originate from the same distribution, which is used to compare two or more independent samples of equal or different sample sizes.

## 5. Conclusions

The PD-L1 status of gastric cancers is the most significant marker to predict the anti-cancer therapy effect. The prominent anti-cancer targets are autophagy-related proteins. During this study, the prevalence of PD-1 and PD-L2 overexpression in IHC PD-L1 positive GCs was found. Immunity modulation is a critical oncogenic process. Cellular signaling in cancers results in the depression of the AKT/mTOR signaling pathway and the activation of transcriptional and growth factors.

This study verified the inner molecular signal indicating a pre-existing resistance in cancers. Hypoxia is an autophagy inducer, and the relationship between immunity and autophagic flux is a well-known adaptation mechanism. Elemental resistance to NACT provokes a reduced response to the treatment. We determined the impact of molecular marker overactivation and NACT on PD-L1 positive GCs. The PD-L1 protein is the driver of immunity in cancers. We also found out the effects of PD-1 and PD-L2 on GCs behavior and the impact this has on the patients’ outcome, resulting in molecular profile modulation and the NACT response.

## Figures and Tables

**Figure 1 ijms-22-11176-f001:**
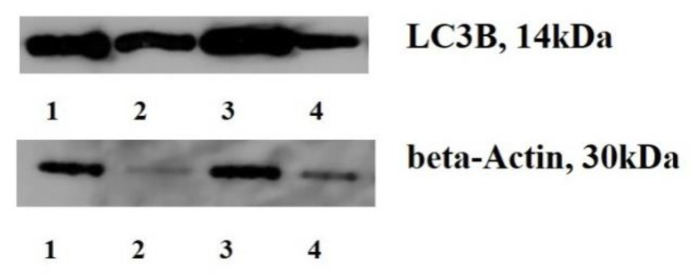
LC3B content in GCs tissues and adjacent non-transformed ones. (Figure 1. Note: 1, 3–cancers, 2,4—non-transformed tissues; LC3B content is a key oncogenic event in GCs development, indicating the autophagy initiation. The modified biological features in cancers after the NACT are responsible for the development of anti-cancer therapy resistance).

**Table 1 ijms-22-11176-t001:** PD, PD-L1, PD-L2 expression in GC tissues depending on IHC PD-L1 status, Me (Q1; Q3).

	PD-L1 Tumor Status	
	PD-L1 Negative, (CPS > 5)	PD-L1 Positive, (CPS < 5)
PD-1	0.69 (0.67; 1.32)	2.67 (0.55; 8.21) *
PD-L1	0.81 (0.31; 1.12)	0.59 (0.31; 0.82)
PD-L2	1.91 (1.07; 2.20)	4.65 (1.70; 18.51) *

Note: *—the significance of differences compared to patients with PD-L1 positive status, *p* < 0.05.

**Table 2 ijms-22-11176-t002:** Molecular markers in GC tissues’ gastric cancer depend on IHC PD-L1 status, Me (Q1; Q3).

	PD-L1 Tumor Status	
	PD-L1 Positive, (CPS < 5)	PD-L1 Negative, (CPS > 5)
AKT/mTOR signaling cascade components and AMPK		
PDK	0.57 (0.51; 1.14)	0.44 (0.19; 0.95)
AKT	1.27 (0.85; 2.73)	1.74 (0.21; 6.91)
c-RAF	1.10 (0.63; 1.50)	1.24 (0.43; 6.75)
GSK-3β	3.13 (0.44; 4.48)	0.68 (0.44; 0.92)
PTEN	1.25 (0.41; 1.57)	1.47 (0.13; 10.36)
mTOR	0.97 (0.69; 1.35)	0.70 (0.09; 1.59)
4EBP1	1.30 (0.94; 2.35)	1.36 (0.15; 2.88)
70s 6 kinase	1.19 (0.72; 2.30)	1.37 (0.72; 4.09)
AMPK	1.16 (0.43; 2.16)	0.79 (0.07; 3.26)
Transcriptional and growth factors		
NF-κBp65	1.34 (0.46; 2.20)	1.38 (0.54; 2.81)
NF-κBp50	0.51 (0.39; 1.36)	7.68 (1.36; 26.70) *
HIF-1	2.30 (0.99; 5.22)	5.16 (0.93; 27.21)
HIF-2	0.83 (0.39; 1.37)	10.25 (0.13; 24.21) *
VEGF	0.50 (0.32; 1.44)	6.97 (0.18; 25.27)
CAIX	1.04 (0.35; 1.90)	0.74 (0.05; 3.66)
VEGFR2	1.04 (0.58; 4.98)	0.59 (0.01; 1.05)

Note: *—the significance of differences compared to patients with PD-L1 positive status, *p* < 0.05.

**Table 3 ijms-22-11176-t003:** Expression of LC3B, mTOR, AMPK, and autophagosome LC3B protein content in gastric tumor tissue depending on PD-L1 tumor status.

	PD-L1 Tumor Status	
	PD-L1 Negative, (CPS > 5)	PD-L1 Positive, (CPS < 5)
LC3B expression, Relative Units	0.57 (0.38; 1.44)	1.99 (0.38; 3.80) *
LC3B protein level, % to the normal tissues	83.00 (55.7; 100.35)	146.65 (126.75; 166.56) *

Note: *—the significance of differences compared to patients with PD-L1 positive status, *p* < 0.05.

**Table 4 ijms-22-11176-t004:** Influence of neoadjuvant therapy on the expression of the AKT/mTOR signaling pathway components and AMPK in gastric tumors.

Indicator, Relative Units	Before NACT	After NACT
AKT/mTOR signaling cascade components and AMPK		
PDF	1.54 (0.35; 6.89)	1.25 (0.06; 9.49)
AKT	0.99 (0.21; 2.21)	1.95 (1.58; 76.00)
c-RAF	6.77 (1.27; 29.60)	15.07 (6.57; 120.76)
GSK-3β	1.95 (0.30; 16.84)	1.12 (0.02; 2.29)
PTEN	2.13 (0.20; 9.78)	7.18 (0.00; 29.82)
mTOR	0.99 (0.19; 1.62)	4.74 (0.15; 6.83)
4EBP1	3.1 (0.45; 56.87)	1.41 (0.48; 15.62) *
70s 6 kinase	3.92 (0.54; 15.40)	8.02 (0.26; 32.66)
AMPK	1.45 (0.11; 7.95)	11.10 (0.07; 15.38)
Transcriptional and growth factors		
NF-κBp65	0.76 (0.23; 2.26)	1.93 (0.78; 10.68)
NF-κBp50	0.60 (0.38; 10.26)	1.33 (0.24; 10.34)
HIF-1	2.02 (0.54; 9.17)	8.12 (0.77; 20.49)
HIF-2	1.37 (0.09; 4.5)	0.89 (0.4; 6.31)
VEGF	0.32 (0.03; 2.13)	2.32 (0.40; 10.75) *
CAIX	0.51 (0.13; 1.46)	2.57 (0.56; 7.04) *
VEGFR2	0.78 (0.20; 5.31)	1.7 (0.33; 2.59)

Note: *—the significance of the differences in comparison with the indicator before treatment, *p* < 0.05.

**Table 5 ijms-22-11176-t005:** Expression of LC3B, PD, PD-L1, PD-L2 in gastric tumor tissue before and after NACT.

Indicator, Relative Units	Before NACT	After NACT
LC3B expression	0.57 (0.28; 1.52)	2.94 (0.85; 3.25) *
PD-1 expression	0.83 (0.31; 2.15)	0.97 (0.66; 1.93)
PD-L1 expression	1.91 (0.77; 7.00)	0.70 (0.05; 1.91)
PD-L2 expression	1.14 (0.20; 3.02)	1.42 (0.56; 11.10)

Note: *—the significance of the differences in comparison with the indicator before treatment, *p* < 0.05.

**Table 6 ijms-22-11176-t006:** Molecular markers in gastric cancers depending on the PD-L1 status after NACT.

	PD-L1 Tumor Status	
	PD-L1 Positive, (CPS < 5)	PD-L1 Negative, (CPS > 5)
AKT/mTOR signaling cascade components and AMPK		
PDK	1.37 (0.60; 2.63)	0.71 (0.00; 1.42)
AKT	2.21 (1.18; 3.32)	0.39 (0.03; 0.75) *
c-RAF	4.95 (1.38; 11.19)	5.76 (2.00; 6.44)
GSK-3β	3.31 (0.44; 6.44)	0.68 (0.44; 0.92)
PTEN	2.33 (1.07; 8.4	5.00 (3.08; 6.92)
mTOR	2.21 (0.61; 4.28)	0.20 (0.00; 0.40) *
4EBP1	2.28 (1.49; 3.04)	1.53 (1.53; 1.54)
70s 6 kinase	1.34 (0.83; 2.77)	1.36 (0.65; 2.07
AMPK	0.71 (0.00; 6.27)	0.56 (0.56; 0.57)
Transcriptional and growth factors		
NF-κBp65	2.80 (2.23; 14.54)	5.45 (0.22; 10.68)
NF-κBp50	0.91 (0.35; 30.36)	4.47 (1.05; 7.90)
HIF-1	3.44 (1.60; 15.81)	92.46 (70.13; 114.80) *
HIF-2	4.45 (2.44; 29.46)	0.75 (0.53; 0.98)
VEGF	0.63 (0.38; 1.68)	5.04 (0.06; 10.75) *
CAIX	2.00 (0.74; 3.58)	28.39 (1.06; 55.72) *
VEGFR2	1.85 (1.14; 14.01)	2.08 (1.58; 2.59)

Note: *—the significance of differences compared to patients with PD-L1 positive status, *p* < 0.05.

**Table 7 ijms-22-11176-t007:** Molecular markers in gastric cancers depending on the PD-L1 status after NACT.

	PD-L1 Tumor Status	
	PD-L1 negative, (CPS > 5)	PD-L1 Positive, (CPS < 5)
LC3B expression, Relative Units	3.25 (2.80; 3.78)	1.75 (0.54; 4.65)
LC3B protein level, % to the normal tissues	83.00 (55.70; 100.35)	146.65 (2.80; 3.78) *
PD-1, Relative Units	5.24 (0,58; 10.40)	71.1 (4.2; 123.34) *
PD-L1, Relative Units	2.39 (0.53; 4.61)	0.88 (0.57; 1.20)
PD-L2, Relative Units	6.52 (0.00; 24.93)	0.45 (0.23; 0.68)

Note: *—the significance of differences compared to patients with PD-L1 positive status, *p* < 0.05.

**Table 8 ijms-22-11176-t008:** Clinical characterization of GC patients.

Characteristics	Participants, n (%)
Gender	
Male	26 (68.4%)
female	12 (31.6%)
Age, years	
20–40	5 (13.3%)
40–60	12 (31.5%)
>60	21 (55.2%)
Tumor stage	
T1-4N0M0	14 (37.0%)
T3-4N1M0	16 (42.0%)
T3-4N2M0	8 (21.0%)
PD-L1 status	
PD-L1+	12 (31.5%)
PD-L1	26 (68.5%)
Response to NACT	
Complete response	3 (12.5%)
Partial response	6 (25%)
Progression	4 (16.7%)
Stabilization	11 (45.8%)

## Data Availability

Not applicable.

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
