# Peer review of "PD-L1 Status in Gastric Cancers, Association with the Transcriptional, Growth Factors, AKT/mTOR Components Change, and Autophagy Initiation"

_ijms, 2021, doi:10.3390/ijms222011176_

Round 1
Reviewer 1 Report
The manuscript entitled "PD-L1 status in gastric cancers, association with the transcriptional, growth factors, AKT/mTOR components change, and autophagy initiation" by Spirina et al. is written in plain English and needs extensive language revision.
Authors describe correlations between PDL1 immunohistochemistry and other molecular markers measured by RT-PCR or Western Blot of autophagy or the AKT/mTOR pathway.
The introduction covers most aspects of the investigation in an appropriate manner including the assumptions concerning autophagy and immunogenicity of tumors.
Results are shown in tables but figures of immunohistochemistry or western blots are missing (at least as supplemental figures).
The discussion part consists almost only of citation and explanations of published articles. Results are not mentioned except for some single sentences, written in a quite generally manner.
In materials and methods tumor classification/stage remains unclear concerning metastasis (N/M): in one sentence, no metastasis is present, in another there is. Data of RT-PCR is presented (I didn't check for correctness and specificity of oligonucleotides). Missing data in respect to used material and methodology for immunohistochemistry (Her2 status as well as PDL1).
Analyzed data is of interest to researchers in the field but major revision is needed:
Introduction could focus more on the aim or hypothesis of the study .
Results should contain more information and figures about IHC and western blots.
Discussion parts need a workover to discuss results and critically look on the selected methods (e.g. RT-PCR of PD1 or PDL1/PDL2 possible to compare with PDL1 expression in other publications?).
In M&M, detailed information about patients, tumor stages in respect to time points (before, after treatment?), age, gender, etc. should be added (might be best to recognize as a table). Authors should add data about mentioned immunohistochemistry.
For all chapters, language revision necessary due to wording, typos, grammar and use of abbreviations without or late clarification as well as unclear sentences (most likely missing words as in "The aim of the study was the assessing the mRNA pattern of PD-1. PD-L1, PD-:2, transcriptional, growth factors, LC3B with the IHC PD-L1 status of gastric patents." (p2,l 97-99)
Author Response
The cteam of authors would like to thank the reviewer for such a thorough analysis. The HER-2 and PD-L1 status detection are the common procedures in gastric cancers therapy.
The Proofreading as made. The Material and methods part of the paper was also corrected. The Discussion part was rewritten.
Reviewer 2 Report
Luidmila Spirina et al.describe in their paper «PD-L1 status in gastric cancers, association with the transcriptional, growth factors, AKT/mTOR components change, and autophagy initiation » the profile of different biological markers in patients with gastric cancer.
General remarks:
The article is very heavy to read. The style of writing should be deeply improved in order to facilitate the reading and the comprehension.
Many typographical errors are present.
example line 279 and 280 GADPH for GAPDH.
Some abbreviations used are not explained until far too late in the text. line 48 NACT does not come until line 70.
Abstract:
The authors mention 38 patients in the study but indicate 12 PD-L1 positive patients and 22 negative patients. This represents only 34 patients. What happened to the 4 missing patients?
Results:
The results are just a succession of tables. The exploitation of these results is only very brief. The authors present in the materials and methods section, the determination of LC3B by western-blotting. I am surprised not to see in the results a figure showing at least one gel.
In conclusion, major revisions must be made before we can consider publishing this work in the International Journal of Molecular Sciences.
Author Response
The team of authors would like to thank the reviewer for such a thorough analysis. All errors and mistakes were corrected, and the abbreviations were also improved. The number of patients was also updated. The WB figures were added to the paper and the supplementary material;
Round 2
Reviewer 1 Report
The revised manuscript entitled "PD-L1 status in gastric cancers, association with the transcriptional, growth factors, AKT/mTOR components change, and autophagy initiation" by Spirina et al. has improved in some ways by this actual changes.
Apart from major changes in the discussion part, only little to no change had been done to the following points:
- Results should contain more information and figures about IHC.
- Discussion parts improves a lot and sufficiently by the changes.
- Introduction could focus more on the aim or hypothesis of the study .
- Detailed data about immunohistochemistry (antibody clone, staining procedure?)
- In M&M, detailed information about patients, tumor stages in respect to time points (before, after treatment?), age, gender, etc. should be added (might be best to recognize as a table).
- No changes were made on language criticisms as:
For all chapters, language revision necessary due to wording, typos, grammar and use of abbreviations without or late clarification (e.g. NACT) as well as unclear sentences (most likely missing words as in "The aim of the study was the assessing the mRNA pattern of PD-1. PD-L1, PD-:2, transcriptional, growth factors, LC3B with the IHC PD-L1 status of gastric patents." (p2,l 97-99)
Author Response
The team of authors would like to thank the referee for a thorough analysis of the article. Information on IHC research has been added. The examination is part of a routine, the generally accepted procedure for diagnosing a disease. Drawings are also standard, and they do not carry any additional information.
The IHC procedure is addedÑŽ The concept was clarified in the introduction.
The clinical data are summarized in Table 8. The English proofreading is made, and the aim of the study is corrected.
Reviewer 2 Report
Thanks to the authors for improving the manuscript in part.
However, the style is still very heavy.
Some abbreviation are still explain to late. line 48 NACT but no meaning that comes only line 70.
Figure 1:
The authors have added a Western blotting figure (figure 1) .If I understand the Appendix B legend correctly there is a comparison between lines 1 and 3 of cancerous tissue versus lines 2 and 4 of non-transformed ( non-Cancerous?) tissue. However, in the analysis of the results, the authors indicate a significant difference between a PD-L1 positive and a PD-L1 negative tumor status. So I don't understand the link with the results. Thank you for clarifying this point which does not convince me.
Author Response
The WB picture shows the results in LC3B content, which later are normalized to the non-altered tissue and the actin level both in cancerous and non-transformed ones. The PD-L1 status in cancers is an indicator in the IHC study. It shows only the number of cells with PD-L1 protein. If the tumor is PD-L1 positive, it has more than 5 cells with the PD-L1 protein among 100 cancer cells.
We found the correlation (only in Analysis) between the IHC PD-L1 status and LC3B expression, and LC3B protein in cancers (after the surgery).
Round 3
Reviewer 2 Report
Thanks to the authors for improving the manuscript. It can now be published.